# Mechanisms and Outcomes of Metabolic Surgery in Type 2 Diabetes

**DOI:** 10.3390/metabo12111134

**Published:** 2022-11-17

**Authors:** Mansor Fazliana, Zubaidah Nor Hanipah

**Affiliations:** 1Nutrition, Metabolism and Cardiovascular Research Centre, Institute for Medical Research, National Institutes of Health, Ministry of Health Malaysia, Selangor 40170, Malaysia; 2Department of Surgery, Faculty of Medicine and Health Sciences, Universiti Putra Malaysia, Selangor 43400, Malaysia

**Keywords:** diabetes, bariatric surgery, metabolic surgery, endocrine, microRNA, gut microbiota, metabolomics

## Abstract

This review is aimed at synthesizing the mechanisms and outcomes of metabolic surgery on the endocrine system, microbiome, metabolomics, and at the molecular level. We review the hormonal, adipokine, microbiota, microRNA, and metabolomic changes in human and animal models following metabolic surgery for the treatment of obesity and diabetes. The most relevant studies in this area over the past 17 years have been considered for this review. In most cases, metabolic procedures, especially those that include intestinal bypass components, showed the remission of type 2 diabetes. This involves a variety of weight-independent mechanisms to improve glucose homeostasis, improving insulin sensitivity and secretion, gut microbiota, and bile acid cross-talk.

## 1. Introduction

Globally rising sedentary lifestyles and obesity have increased, resulting in the tremendous burden of type 2 diabetes (T2D) worldwide. Bay et al. reported in a study conducted in the US that approximately 85% of patients with T2D are either overweight or obese [1]. The literature has shown that weight loss surgery (i.e., bariatric or metabolic surgery) has been proven to be effective and provides good long-term glycemic control in patients with obesity and T2D. Metabolic surgery is now considered a well-supported treatment for T2D in patients with obesity and is endorsed by international diabetes and medical organizations [2].

Metabolic surgery exerts its physiological benefits by mediating intestinal physiology, bile acid metabolism, incretin hormone secretion, neuronal signaling, and microbiome changes. Understanding the mechanisms of metabolic procedures and their outcomes on diabetes will ensure optimal treatment and disease prevention strategies in T2D patients. Although the benefits of metabolic surgery for obesity-related comorbidities are well-established, evidence of its molecular and metabolomic effects is still limited.

This review article summarizes the effects of metabolic surgery on the endocrine system, gut microbiota, microRNA levels, and metabolomics for the treatment of obesity and diabetes, both in clinical trials as well as in animal models.

## 2. Overview of Bariatric/Metabolic Surgery

Bariatric surgery was initially performed for weight loss (baros = weight) [3]. It was initially applied to the treatment of morbid obesity, as documented in 1954 [4]. The procedure was originally designed to achieve and sustain weight loss, and it was consequently noted to induce improvements in glucose regulation [5,6]. Bariatric surgical procedures included jejunoileal bypass (JIB), Roux-en-Y gastric bypass (RYGB), sleeve gastrectomy (SG), biliopancreatic diversion (BPD), vertical-banded gastroplasty (VBG) or its familiar duodenal switch (DS), and adjustable gastric banding (AGB) [7]. Currently, RYGB and SG are two of the most popular types of bariatric surgeries that are applied in the treatment of obesity and T2D worldwide [8]. RYGB and SG have very different anatomical structures, but to varying degrees both surgeries are effective at inducing weight loss and T2D remission [9].

The surgery itself has led to the mechanical hypothesis that bariatric operations generally promote weight loss by restricting the stomach size and/or bypassing some of the intestines to cause nutrient malabsorption. However, the remission of T2D has been observed after gastric bypass before the occurrence of significant weight reduction. It shows that the glucose improvement resulting from metabolic operations occurs not only because of weight loss but also due to weight-independent factors, such as gut hormones, bile acids, gut microbiota, the nervous system, and other potential underlying mechanisms [10,11,12,13]. Short- and long-term clinical studies have shown that bariatric surgery results in substantial weight loss and either the improvement or remission of T2D [13,14,15]. As clinical data accumulated, some versions of these procedures were ultimately abandoned because of complications related to the surgery.

Many studies have shown that bariatric surgery, which is now known as metabolic surgery, improves glucose homeostasis. Bariatric procedures, especially those with intestinal bypass components, have been shown to lead to the remission in T2D. In a recent 2-year follow-up study, the complete remission of T2D was achieved in nearly half of patients two years after metabolic surgery [14]. A large cohort with prospective randomized interventional data demonstrated that metabolic surgery improves glycemic control, diabetes-related comorbidities, and weight loss to a greater extent than lifestyle/medical intervention for up to 3 years postsurgery [15]. Studies assessing the impact of metabolic surgery on the control of diabetes are summarized in Table 1.

The principal determinants of glucose homeostasis, which include insulin secretion, insulin sensitivity, and insulin-independent glucose disposal, are affected by RYGB, VBG, and BPD [16]. A comprehensive understanding of these effects helped to optimize surgical techniques and devices to provide the maximum antidiabetes impact.

**Table 1 metabolites-12-01134-t001:** Summary of studies assessing the impact of metabolic surgery on the control of diabetes.

Study	Intervention/ Surgery Type	Subjects (n)	Results	Outcomes
Vrakopoulou et al. (2021) [17]	SG OAGB	morbidly obese T2D patients SG (n = 28) OAGB (n = 25)	SG-10 patients (35.7%) achieved diabetes remission. OAGB-22 patients (88%) remained off antidiabetic agents.	OAGB was more effective in improving glycemic control and %EWL with an almost immediate resolution of diabetes as well as long-term weight loss.
Moriconi et al. (2021) [18]	RYGB	RYGB: 88 obese patients Medical therapy (MT): 25 obese patients 10-year follow-up	Body mass index (BMI), fasting glucose, and hemoglobin A1c (HbA1c) ↓ in RYGB than MT patients.	RYGB significantly reduced and sustained glycated hemoglobin (HbA1c) levels compared to medical therapy at a ten-year follow-up. Weight loss impacted the short-term remission of T2D but had a marginal role in long-term relapse.
Imtiaz et al. (2021) [19]	bariatric surgery	Randomized controlled trials (RCTs) among T2D patients 3674 with surgery 1335 without surgery	HbA1C ↓ Improvements were sustained for at least 6 years.	Bariatric surgery lowered A1c in real-world clinical care.
Purnell et al. (2021) [20]	RYGB LAGB	2256 participants, 827 with T2D and severe obesity	Diabetes remission occurred in 57% (46% complete, 11% partial) after RYGB and 22.5% (16.9% complete, 5.6% partial) after LAGB. younger participants, shorter diabetes duration: remission↑	Durable long-term diabetes remission following bariatric surgery was more likely when performed soon after diagnosis, when the diabetes medication burden was low and beta-cell function was preserved.
Huang et al. (2021) [21]	single-anastomosis gastric bypass	1999 patients from the Asian Diabetes Surgery Summit 1 year	Weight↓ BMI ↓ Blood glucose ↓ HbA1c ↓ Lipid ↓ Blood pressure ↓	Metabolic surgery remarkably improved body weight, T2D, and other metabolic disorders in Asian patients. However, the efficacy of individual procedures varied substantially.
Stenberg et al. (2020) [22]	RYGB	742 patients at baseline and 2 years postsurgery	Insulin homeostasis and glucometabolic control were improved and sustained.	Positive results were observed in diabetics as well as prediabetics and nondiabetic obese patients, and this improvement was sustained for 2 years after surgery.
Katsogiannos et al. (2020) [23]	RYGB	19 obese T2D patients	parasympathetic nerve activity ↑ morning cortisol ↓ incretin ↑ glucagon responses ↑	Neurohormonal mechanisms can contribute to the rapid improvement in insulin resistance and glycemia following RYGB in T2D.

## 3. Mechanisms Involved in the Postoperative Weight and Metabolic Changes

Multiple mechanisms are responsible for postoperative weight and metabolic changes. Obesity and T2D are biologically linked, with obesity being a primary driver of insulin resistance. It is also implicated in β-cell decompensation [24]. Long-term weight loss can dramatically improve the lives of people with established T2D [25,26,27,28]. A negative energy balance and the weight loss it produces have powerful effects on many physiologic factors that could lead to the remission of the T2D disease process [29]. The association between weight loss and its powerful effects in improving these comorbidities is only now being explored. We discuss the mechanisms that are responsible for postoperative weight and metabolic changes. These involve some degree of overlap involving the endocrine, digestive, nervous, and immune systems and metabolomic changes as well as changes on the molecular level. The hormonal and adipokine changes after metabolic surgery include adiponectin, ghrelin, leptin, glucagon-like peptide, secretin, and oxyntomodulin [30,31,32].

### 3.1. Hormonal Changes

Metabolic surgery improves insulin secretion and sensitivity in patients with T2D, but the effect on patients with normal glucose tolerance or prediabetes (pre-DM) needs further understanding. Stenberg et al. [22] followed 742 patients after RYGB. The glucometabolic control and insulin homeostasis improved in all these patients, and these results were sustained for 2 years. In another study following RYGB in T2D patients, there was a significant improvement in glycemia and insulin resistance, which could be due to neurohormonal mechanisms [33].

The absorption of glucose and protein was greatly accelerated after RYGB but was only modestly accelerated after SG. Insulin, glucagon-like peptide-1 (GLP-1), cholecystokinin (CCK), and peptide YY (PYY) are appetite inhibitors, and their secretions also differed markedly between the procedures [34]. The mechanisms underlying the benefits of metabolic surgery likely involve the bile acid signaling pathway [35].

Sixty-nine subjects who underwent RYGB were compared with matched controls who had never been obese in a 2-year longitudinal study. The RYGB group showed slightly increased HOMA-IR insulin sensitivity, a beneficial body composition, higher insulin clearance, and lower atherogenic lipid and lipoprotein levels as well as benign adipocyte morphology (*p* < 0.0001 for these parameters). This partly explained why long-term metabolic complications were protected by RYGB [36]. A cross-sectional study of 36 T2D patients was performed to study the absorption rates of glucose and protein as well as the profiles of gastro-entero-pancreatic hormones after metabolic surgery [34]. Metabolic surgery was performed in 24 patients (LSG, n = 12; RYGB, n = 12) and compared to a control group (n = 12). They received continuous infusions of stable isotopes of glucose, glycerol, phenylalanine, tyrosine, and urea before and during a mixed meal containing labeled glucose and intrinsically phenylalanine-labeled caseinate. This study showed that the systemic appearance of ingested glucose was faster after RYGB and SG vs. controls, while the peak glucose appearance rate was 64% higher after RYGB and 23% higher after SG (both *p* < 0.05). The postprandial glucose and protein absorption and gastro-entero-pancreatic hormone secretions differed after SG and RYGB. RYGB was characterized by the accelerated absorption of glucose and amino acids, whereas protein metabolism after SG did not differ significantly from controls, suggesting that different mechanisms explain the improved glycemic control and weight loss after these surgical procedures [34].

In a long-term study involving 163 patients with T2D, metabolic surgery was able to reduce obesity-related chronic low-grade inflammation. The acute-phase reactants C-reactive proteins (CRP, *p* < 0.001) and high-sensitivity CRP (hs-CRP, *p* < 0.001) are commonly used to monitor inflammation and are strongly associated with metabolic syndrome, atherosclerotic cardiovascular disease, and T2D. These markers were significantly reduced for up to 4 years after surgical interventions. The improvement was related to the change in BMI and the remission of T2D in the long term [37].

Ghrelin and GLP-1 have a very important role in this mechanism, which will be discussed further below.

a.Ghrelin

The main ghrelin functions are targeted at appetite, metabolism, and adiposity. Acylated and deacyl ghrelin are the two main circulating isoforms of this hormone [38]. These isoforms consist of the main elements that are involved in the amelioration of nonalcoholic fatty-liver disease (NAFLD) after bariatric surgery [39]. In a ten-week diabetic rat study, RYGB and SG decreased leptin and ghrelin levels [40]. Ghrelin affects carbohydrate and lipid metabolism in obese patients. After RYGB and LSG surgeries, ghrelin was associated with elevated plasma levels of insulin, leptin, and glucagon [41].

b.Glucagon-like peptide-1 (GLP-1)

RYGB leads to profound changes in the secretion of gut hormones, with effects on food intake and appetite as well as metabolism [29]. Several studies highlighted the important role of GLP-1 in achieving glycemic control. GLP-1 is elevated after RYGB. Bile acid plasma concentrations were increased after RYGB. In this case, bile acids may act as molecular enhancers of GLP-1 secretion through the activation of TGR5 receptors [42]. The improved glucose tolerance was due to a negative energy balance and the resulting weight loss. In the beginning, these improve hepatic insulin sensitivity, and they later improve peripheral insulin sensitivity. Next, in combination with elevated postprandial insulin secretion, insulin sensitivity is elicited, particularly by magnified GLP-1 responses [29]. Additionally, RYGB causes a weight-loss-independent postprandial insulin secretion, which contributes to the improvement in glycemic control. This action is associated with a ~10-fold increment in the concentrations of the incretin hormone GLP-1 in plasma [43]. 

A meta-analysis by Jirapinyo et al. of 24 studies involving 368 patients concluded that GLP-1 fasting levels remain unchanged, but the levels increase after RYGB. Interestingly, shorter a Roux limb length is associated with a greater increase in postprandial GLP-1, which may improve glycemic control [29].

Using a model of gastrectomy in lean mice, Larraufie et al. showed that after bariatric surgery GLP-1 is an enhancement factor of insulin secretion, which arises from fast nutrient delivery to the distal gut [44]. In an animal study mentioned earlier, RYGB and SG increased GLP-1 levels [40], while another study showed the complete remission of T2D, which was significantly associated with higher GLP-1 levels [14].

An animal study revealed that the microbiota changes by RYGB play a key role in postsurgical weight loss [45]. Besides bile acids, short-chain fatty acids (SCFAs) were also reported to modify the secretion of GLP-1 [46]. Likely due to RYGB at the point of inclusion of the bypass of the proximal intestine, elevated GLP-1 secretion, altered gut microbiota, and increased SCFA in the gut were reported after pancreatoduodenectomy [47]. The effect of gut microbiota composition on lipid profiles was analyzed after RYGB. It was found that SCFA-producing bacteria promote healthy lipid homeostasis, while the presence of LPS-producing bacteria such *Escherichia-Shigella* may contribute to the development of atherogenic dyslipidemia [48]. Microbiota changes in metabolic surgery are discussed further in Section 3.3.

### 3.2. Adipokine Changes 

Adipose tissue dysfunctionality could be caused by excessive visceral fat accumulation. This contributes significantly to the onset of obesity-related comorbidities [49]. Higher visceral fat at follow-up exams was significantly associated with reduced remission, which increased the incidence of diabetes, dyslipidemia, and hypertension [50]. Hepatic, adipose, and skeletal muscle tissues are the crucial endocrine organs that produce hepatokines, adipokines, and myokines. They are biomarkers that can be beneficial or detrimental to an organism and act through the autocrine, endocrine, and paracrine pathways [51].

Among the varieties of adipokines produced by adipose tissue are cytokines (TNF-α, TGFβ, IL-R1a, IL-6, and IL-10), chemokines (MCP-1, CCL2, CCL5, macrophage inflammatory protein-2 (MIP-2), IL-8/CXCL8, and IFNγ-inducible protein 10/ CXCL10), acute-phase reactants (haptoglobin, serum amyloid A, C-reactive protein, and plasminogen activator inhibitor 1), damage-associated molecular pattern molecules (calprotectin, DAMPs, HMGB1, tenascin C, and heat shock protein 72), and proinflammatory (leptin, osteopontin, resistin, WNT5A, and chemerin) as well as an anti-inflammatory (ghrelin, adiponectin, SRFP5, lipocalin-2, and omentin) adipokines [30]. Obesity is associated with a changed secretion of adipokines that translates into increased cardiovascular risk in patients. These obese patients have an excess of dysfunctional adiposity [49]. 

The functions of specific adipokines and their effects on metabolic surgery are further discussed.

a.Leptin

A product of the obesity gene, leptin takes part in the regulation of body weight by controlling food intake and energy expenditure [52]. Leptin activates a balanced effect on blood pressure control in the healthy state by modulating the endothelial release of nitric oxide as well as sympathetic activity-dependent vasoconstriction and angiotensin II-dependent vasoconstriction [30,53]. In obesity, hyperleptinemia may emerge as a compensatory mechanism to control leptin resistance. This is due to obesity, which activates an organ-specific leptin-resistant state [54]. Šebunova N. et al. analyzed 30 obese bariatric patients and found a remarkable decrease in leptin levels [55]. 

b.Adiponectin

Adiponectin is only expressed in adipose tissue, which is detectable in plasma. Plasma adiponectin concentrations are reduced in obese patients. Adiponectin exerts anti-inflammatory actions as well as increases insulin sensitivity [56]. Metabolic syndrome and insulin resistance in humans are best predicted by high-molecular-weight adiponectin [57].

Moreover, the adiponectin/leptin ratio has been recommended as a marker of adipose tissue dysfunction. The ratio correlates with insulin resistance more closely than a surrogate of insulin resistance such as the HOMA index, adiponectin, or leptin alone [58]. Metabolic surgery resulted in weight loss and activated a constant decrease in leptin levels and a parallel increase in adiponectin plasma levels [55,59]. 

c.Resistin and visfatin

Adipose-tissue-resident macrophages secrete resistin, which is a polypeptide. Its concentrations are elevated in obesity. This is because the pathophysiology of inflammation-induced insulin resistance in macrophages is regulated by circulating resistin levels [60]. Prospective case-control studies proved an association between an increased risk of developing T2D and subjects with increased resistin levels at baseline. These levels decreased after bariatric surgery [61,62]. 

Another adipocytokine secreted by adipocytes, inflamed endothelial tissue, and macrophages is visfatin. It increases obesity, insulin resistance, and T2D. Visfatin acts as a proinflammatory intermediary has an important role in vascular inflammation pathogenesis in obesity and T2D, and contributes to atherosclerotic plaque instability. An improvement in insulin resistance and diabetes was reflected in T2D patients who underwent RYGB when the visfatin serum level was decreased [62,63].

d.Omentin-1 and apelin

Omentin-1, also known as intelectin-1, is an adipokine that is primarily secreted from visceral adipose tissue and consists of 313 amino acids, but it is also expressed in the heart, placenta, and ovaries [64,65]. It is an anti-inflammatory adipokine [66] that is expressed in omental, epicardial, and perivascular adipose tissue [67]. In obesity, the omentin-1 level, which is the major circulating form, is reduced. It is also inversely correlated to waist circumference, BMI, and metabolic syndrome biomarkers [68]. Its expression is reduced in obesity [69]. During diet-induced weight loss, omentin-1 levels usually elevate over time, which is evidence of a link between omentin and obesity [70]. 

The circulating omentin levels and the related mRNA expression in visceral adipose tissue are distinct in different types of diabetes [68]. There might be a protective action of omentin in metabolic disorders [71]. Omentin reduced insulin resistance in Goto-Kakizaki rats fed a high-fat diet without affecting their lipid profile [72]. Interestingly, a systematic review and meta-analysis found that serum omentin levels are significantly lower in impaired glucose tolerance and T2DM patients but not in type 1 diabetes (T1DM) [73]. 

In a study, most postbariatric patients displayed an elevation of omentin-1 levels in the immediate postoperative period. This condition even occurred before the induction of weight loss. The increment in omentin-1 levels was even maintained for one year after the bariatric intervention [74]. 

Another novel adipokine, apelin, has a crucial role in the pathogenesis of insulin resistance as well as T2D. Apelin is secreted from white adipose tissue and is associated with various functions, including food intake and insulin sensitivity [75,76]. The level of apelin in obese patients with T2D is significantly increased compared to healthy people [77]. In obesity and diabetes, insulin could control apelin [78]. 

Long-term apelin treatment in insulin-resistant obese mice has proven, valuable effects on both glucose and lipid metabolism [79]. During a hyperinsulinemic–euglycemic clamp in nondiabetic human volunteers, apelin perfusion markedly improved insulin sensitivity without causing side effects [80].

### 3.3. Role of Gut Microbiota, Bile Acids, and Their Cross-Talk

One of the important mechanisms for the improvement in T2D after metabolic surgery is the change in the gut microbiota. The gut microbiota structure has an important role in the improvement in islet β-cell function and the hypoglycemic effect. Body weight gain, blood serum lipids, and fasting blood glucose are effectively decreased by the modified jejunoileal bypass [81]. This potential therapeutic strategy for T2D is explained by the jejunoileal bypass, which has modified and improved the gut microbiota composition [81]. Additionally, insulin resistance, islet β-cell function, and glucose tolerance were significantly improved. In a T2D rat experimental model, it was suggested that the islet β-cell function might be contributed by amino acid metabolism [82].

More evidence suggests that the gut microbiota is associated with the development of several metabolic disorders. Bile acids and nuclear bile acid receptor (FXR) signaling are important for the metabolic benefits of metabolic surgery. Furthermore, the microbiota–bile acid interactions play a role in the mechanisms underlying the effects of metabolic surgery [83]. Metabolic surgery can change the intestinal microorganism pattern in response to gastric restriction or the rearrangement of the intestinal tract. Altered nutrient presentation resulting from an incompletely digested diet that enters the downstream gut after different surgical procedures could alter the gut environment and affect the composition of the intestinal bacteria [84]. In a long-term effect study, it was found that the gut microbiota may play a direct role in the reduction in adiposity after metabolic surgery, as demonstrated in the rodent model. The surgically altered microbiota in recipient mice promoted reduced fat deposition [45].

The transfer of the gut microbiota from RYGB-treated mice to sham-operated, germ-free mice resulted in weight loss and decreased fat mass in the recipient animals. This was potentially due to altered microbial production of short-chain fatty acids. These studies show that changes in the gut microbiota contribute to reduced host weight and adiposity after RYGB surgery [85].

The combination of physiology and the computational modeling of microbiota metabolism would motivate researchers to optimize the diagnosis and treatment of T2D patients in a personalized way [86]. This could be applied in metabolic surgery as an option for the treatments.

In addition to inducing inflammation, the gut microbiota plays an important function in modulating bile acids and including their biosynthesis and biotransformation. Cholic acid, one of the bile acids, regulates the gut microbiota composition in rats, inducing changes similar to those induced by high-fat diets. This explains the relationship between the gastrointestinal microbiota composition and metabolic diseases [87]. Altered bile acid levels and compositions may contribute to improved glucose and lipid metabolism in patients who have had gastric bypass [88]. Postsurgical alterations occur in intestinal anatomy, satiety, the secretion of gastrointestinal peptides, neural signaling, and nutrient absorption. These aspects contribute to weight loss and the associated improvements in systemic metabolism. Additionally, an increase in serum bile acid levels also contributes to improved carbohydrate, lipid, and energy metabolism after bariatric surgery. Bile acids and nuclear bile acid receptor FXR signaling were found to be important molecular underpinnings for the beneficial effects of this weight loss surgery [89]. 

### 3.4. Molecular Changes—MicroRNA

MicroRNAs (miRNAs) are expressed in various organs [90]. miRNAs are short pieces of RNA that are recognized as key gene expression regulators and have main roles in the regulation of many biological and pathological processes, including T2D [91]. Owing to their stability and practicality in noninvasive collection methods, circulating miRNAs could be used as biomarkers. This is being explored in a wide range of pathologies, including diabetes and cancer. Furthermore, their levels can be measured by quantitative RT-PCR, which is straightforward, fast, and specific, with sensitive detection and quantification [92].

The differential expression of circulating miRNAs before and after various dietary and bariatric surgery interventions has been reported in a few studies, identifying several weight-loss-related candidate biomarkers. A range of dysregulated miRNA target pathways has also been identified. This is to understand the underlying obesity and obesity-related metabolic disease pathophysiological mechanisms [93].

Dysfunctional adipose tissue is extensively associated with T2D development and is the major source of circulating miRNAs. A specific miRNA, miR-122, which is distinctively found between visceral and subcutaneous adipose tissues, was found in a study [94]. It is involved in weight homeostasis as well as numerous metabolic processes [95]. The ratio of miR-122 between subcutaneous and visceral adipose tissues correlates with the outcome of bariatric surgery [94]. 

miRNA disorders have been demonstrated in various studies involving β-cell development, insulin production, insulin secretion, insulin sensitivity, insulin resistance, and insulin signaling pathways and finally lead to the development of T2D [96]. These findings support the possible role of miR-33 in monitoring prediabetes onset and progression.

The metabolic responses in the early stages following weight loss after bariatric surgery are evident. These observations correspond with an improvement in diabetes. Seven microRNAs (let-7i-5p, let-7f-5p, miR-7-5p, miR-15b-5p, miR-205-5p, miR-320c, and miR-335-5p) showed significant changes 3 weeks after RYGB surgery among 29 patients with severe obesity and T2D. Altered miRNA functional pathways were associated with liver-, diabetes-, and pituitary-related diseases. Following bariatric surgery, the miRNA expression in natural killer cells and vital intestinal pathology imply mechanistic functions in early diabetes responses [97].

In a diet and cardiovascular study cohort, the baseline levels of miR-223-3p were found to be significantly related to insulin resistance in adipose tissue. Both miR-223-3p-secreting preadipocytes and miR-223-3p-secreting adipocytes caused alterations in the circulating levels. This suggested that inflammation enhances the intracellular accumulation of miR-223-3p. This possibly contributes to preadipocyte dysfunction and body metabolic dysregulation [98].

The usefulness of identifying genetic differences between high- and low-weight-loss groups after bariatric surgery by identifying specific serum miRNA has been demonstrated [95]. The miRNA profile in the serum of plasma is deregulated in the pre-DM state before the development of observable T2D. Undoubtedly, compared with controls, individuals with T2D or pre-DM have a differential profile of circulating miRNAs [99]. Interestingly, a list of miRNAs has been identified in nondiabetic healthy individuals who proceeded to develop pre-DM or T2D [100]. Additionally, multiple circulating miRNA plasma concentrations of healthy individuals have been identified as being markedly different between T2D patients and pre-DM individuals [101,102]. Deregulated plasma levels of miR-15a, miR-30a-5p, miR-150, and miR-375 were detected years before the onset of T2D and pre-DM and could be utilized to assess the risk of developing the disease. This may improve the prediction and prevention among high-risk individuals for T2D [103]. Eikelis et al. also demonstrated that the plasma levels of miR-9, miR-28-3p, miR29a, miR-103, miR-30a-5p, and miR-150 are powerful predictive biomarkers that can discriminate between incident-T2D and non-T2D patients. A potential tool for the early detection of T2D has been developed. It is a multiparameter diagnostic model consisting of miR-148b, miR-223, miR-130a, and miR-19a [104]. miR-132 (mir-132) is an important regulator of liver homeostasis and lipid metabolism. In the same experimental model, an association between miR-132 and the markers of metabolic and cardiovascular disease was found [104]. Part of the blood biochemical changes of diabetes reversal was formed during the resolution of diabetes after bariatric surgery. This process occurred through the miRNA–gene interactions in the pancreatic islet, which is a novel mechanism [105].

Soon, informed decisions about surgery could be facilitated by these miRNAs. These potential miRNA biomarkers could also provide targets for future treatments by opening new genetic pathways that illustrate the pathophysiology of obesity. Table 2 presents differentially expressed miRNAs with their roles/targets after surgery.

### 3.5. Metabolomics 

As a relatively young scientific discipline, metabolomics shows great potential for the comprehensive study of the metabolome’s dynamic alterations. Liquid chromatography–mass spectrometry (LC-MS) and nuclear magnetic resonance (NMR) are the most frequently used techniques to study the main effects of RYGB or SG. NMR can uniquely identify and simultaneously quantify a wide range of analyses of amino acids, carbohydrates, vitamins, thiols, and peptides as well as nucleotides and nucleosides. The LC-MS technique has become a powerful tool for the analysis of the polar metabolites in a complex sample [115,116].

In an exploratory study among 17 diabetic and nondiabetic obese patients undergoing bariatric surgery, untargeted metabolomic profiling in subcutaneous adipose tissue was performed. However, among the 421 metabolites that were identified and analyzed, there were no significant differences between those patients [117]. The dysregulation of lipids and amino acids has been associated with insulin resistance and other pathophysiological processes of T2DM. This may be due to obesity, which may influence subcutaneous adipose tissue metabolism, masking T2DM-dependent dysregulation [118].

In another post-RYGB study, alterations in basal metabolism among overweight diabetic subjects were demonstrated by NMR metabolomic profiling. These changes were associated with energy homeostasis, alterations in lipid metabolism, and decreased branched-chain amino acids [119]. 

Most methods for the screening and prevention of T2D rely on prediabetic individuals that are already showing a steady decrease in insulin sensitivity. It is important to develop biomarker trajectory models that can accurately complement the existing individual risk assessment methods because these methods may not be as effective as those developed to counter the disease [120].

In a 5-year diabetes remission study, metabolites in the branched-chain amino acid (BCAA) and trimethylamine-N-oxide (TMAO) microbiome-related pathways were found to be predictive of T2D remission and weight loss amounts in severely obese individuals [121]. These metabolites can potentially be used in the clinical management of T2DM patients undergoing bariatric surgery. Additionally, the baseline levels of tryptophan, bilirubin, and indoxyl sulfate measured before surgery as well as the levels of FFA 16:0, FFA 18:3, FFA 17:2, and hippuric acid measured 6 months after surgery best predicted the suitability and efficacy of RYGB for patients with T2DM [122]. A summary of the metabolomic profiles associated with metabolic surgery is presented in Table 3.

## 4. Conclusions

Multidisciplinary weight management approaches are emerging as feasible and potentially cost-effective treatment options for patients with overweight/obesity and diabetes. Metabolic surgery is a potential and sustainable treatment that can modify a patient’s physiology and glucose regulation mechanism. In most cases, metabolic procedures show the remission of T2D. This involves a variety of weight-independent mechanisms to improve glucose homeostasis, improving insulin sensitivity and secretion. 

The potent metabolic effects of metabolic surgery are not only shown by improved obesity, glucose tolerance, and insulin sensitivity. Recent studies show that microbiota–bile acid interactions play a role in the mechanisms underlying the effects of metabolic surgery. Furthermore, metabolic improvement could be monitored by miRNA levels. 

The overall message for clinicians is that metabolic surgery can be an excellent and life-extending option for some patients with T2D. However, these procedures are associated with perioperative risks and other aspects that are not discussed in this review, such as permanently changing patients’ relationship with food, sometimes involving psychiatric disorders and requiring lifelong diet support and medical monitoring. These aspects need further discussion in their respective areas.

## Figures and Tables

**Table 2 metabolites-12-01134-t002:** Differentially expressed miRNAs with their roles/targets after surgery.

Study	Intervention	Population	Source	Regulated miRNAs	Role/Target
Sangiao-Alvarellos et al. (2020) [106]	Bariatric surgery	155 obese patients 47 CTRL	Serum/plasma	miR-122 miR-885-5 p miR-192	Regulation of hepatic biochemical processes
Cereijo et al. (2020) [107]	Bariatric surgery	26 obese patients	Serum	miR-92a	Glucose homeostasis
Macartney-Coxson et al. (2020) [95]	RYGB	15 obese women	SAT VAT	SAT: miR-23a-5p, miR-27a-5p, miR-200c-3p, miR-223-3p, miR-1246, miR-24-2-5p, miR-128, miR-421, miR-3178, miR-1224-5p, miR-221, miR-22, miR-762 (↓) VAT: miR-223-3p (↓)	Inflammation, glucose uptake
Doyon et al. (2020) [95]	LSG RYGP	20 obese patients	Serum	has-miR-375 hsa-miR-126-3 p hsa-miR-663 a hsa-miR-30 c-5 p hsa-miR-100-5 p hsa-miR-27 a-3 p hsa-miR-590-5 p	Fatty acid biosynthesis, obesity, adipocyte proliferation, T2D
Bae et al. (2019) [94]	LSG (n = 2) RYGB (n = 14)	16 obese patients 18 CTRL	Serum exosomes	miR-424-5p	Biomarker of weight loss
Liao et al. (2018) [94]	LSG	20 obese patients 8 CTRL	SAT VAT	VAT: miR-122 (↑)	PPAR-γ
Atkin et al. (2018) [97]	RYGB	29 T2D patients	Plasma	miR-7-5p, let-7f-5p, miR-15b-5p, miR-320c, miR-205-5p, miR-335-5p (↑) let-7i-5p (↓)	Inflammation, adipocyte proliferation, ß-cell function, thyroid and pituitary function
Nunez-Lopez et al. (2017) [108]	RYGB	22 obese patients	Plasma	miR-15a (↑) miR-34a, miR-122 (↓)	Biomarkers of weight loss/glucose metabolism
Hubal et al. (2017) [109]	RYGB	6 obese women	Plasma and serum adipocyte-derived exosomes	let-7a-5p, miR-16-5p	Insulin signaling
Nardelli et al. (2017) [110]	LAGB	3 obese patients 2 CTRL	SAT	miR-519d, miR-299-5p, miR-212, miR-671-3p (↓) miR-370, miR-487a (↑)	PPAR-α (miR-519d)
Kurylowicz et al. (2016) [111]	Bariatric surgery	20 obese patients 7 CTRL	SAT	miR-146b-3p, miR-146b-5p, miR-223-3p, miR-223-5p, miR-941 (↑)	BMPR2, FOXP1, IGF1R
Ortega et al. (2015) [110]	RYGB	16 obese patients	SAT	miR-155, miR-221, miR-130b (↓)	Inflammation
Ortega et al. (2015) [112]	RYGB	9 obese women	SAT	miR-19a/b, miR-146a/b, miR-155, miR-193b, miR-221, miR-222, miR-223, miR-376c, miR-411 (↓)	Glucose uptake, lipid metabolism, energy homeostasis
Ortega et al. (2013) [113]	RYGB	6 obese patients	Plasma	miR-221 and miR-199a-3p (↑) miR-16-1, miR-122, miR-140-5p, miR-193a-5p (↓)	–
Hulsmans et al. (2012) [114]	RYGB	9 obese patients 6 CTRL	PBMC	miR-181 (↑)	TLR-NFkB pathway

CTRL, control subjects; RYGP, Roux-en-Y gastric bypass; PBMC, peripheral blood mononuclear cells; T2D, type 2 diabetes; SAT, subcutaneous adipose tissue; VAT, visceral adipose tissue; LSG, laparoscopic sleeve gastrectomy; BMPR2, bone morphogenic protein receptor 2; FOXP, forkhead box protein P1; IGF1R, insulin-like growth factor receptor 1; LAGB, laparoscopic adjustable gastric banding; ↑ indicates increased expression level; ↓ indicates reduced expression level.

**Table 3 metabolites-12-01134-t003:** Metabolomic profiles of metabolic surgery.

Study	Intervention/Surgery Type	Population	Follow-Up Period	Role/Target Metabolites	Outcomes
Zheng et al. (2021) [123]	RYGB	38 individuals with T2D	12 months	Hyocholic acid (HCA)	Serum HCA levels increased in the patients after RYGB (*p* < 0.05). HCA species play critical roles in regulating glucose homeostasis and are protective against the development of T2DM in humans.
Ha et al. (2020) [124]	SG RYGB	24 individuals with T2D	3 months	Amino acid metabolites (AAMs): L-DOPA and 3-HAA	The prognostic performances of L-DOPA (AUROC = 0.97; 95% CI, 0.91 to 1.00) and 3-HAA (AUROC = 0.86; 95% CI, 0.63 to 1.00). AAMs are superior for predicting T2D remission postoperatively compared with existing prediction models.
Zhao et al. (2017) [125]	RYGB	419 individuals 38 obese individuals with T2D after RYGB 381 T2D and CTRL with overweight or obesity	12 months after RYGB	Targeted: branched-chain amino acids (BCAAs), aromatic amino acids (AAAs), and acylcarnitines	Higher baseline stearic acid/palmitic acid ratio (S/P) was associated with a greater probability of diabetes remission after RYGB (odds ratio, 2.16 (95% CI 1.10–4.26)) and may serve as a diagnostic marker in preoperative patient assessment.
Narath et al. (2016) [125]	RYGB	44 obese, including 24 patients with T2D at baseline and 9 with diabetes remission	12 months	Untargeted: 36 metabolites that are known markers for cardiovascular risk	Trimethylamine-N-oxide, alanine, phenylalanine, and indoxyl-sulfate are four metabolites that significantly decline in patients with diabetes remission compared to patients without diabetes remission (sarcosine: *p* = 0.031, pyroglutamic acid: *p* = 0.044, alanine: *p* = 0.005, and leucyl-proline: *p* = 0.049).
Lopes et al. (2016) [126]	RYGB	10 obese individuals with T2DM	12 months	Untargeted: metabolic and lipoprotein profiles and fatty acid profile	Glucose levels decreased significantly after RYGB (from 159.8 ± 61.4 to 100.0 ± 22.9 mg/dL), demonstrating T2D remission (*p* < 0.05). Lower levels of metabolic profile: lactate, alanine, and branched-chain amino acids The VLDL, LDL, N-acetyl-glycoproteins, and unsaturated lipid levels decreased, but phosphatidylcholine and high-density lipoprotein increased after RYGB.
Sarosiek et al. (2016) [126]	SG or full GB	15 patients: gastric sleeve with T2D (5), gastric sleeve without T2D (5), and gastric bypass with T2D (5)	28 days	Nontargeted global metabolomics: glucose and lipid metabolism, histidine, and its metabolites	A total of 62 compounds were significantly different postsurgery compared to baseline (*p* < 0.05). Significant improvement in fat mobilization and oxidation (*p* < 0.05) and liver function (*p* < 0.05) after surgery.
Nemati et al. (2016) [127]	LGB (Roux), LSG	38 obese individuals with T2D: GBP (11) SG (14) VLCD (13)	3 days	Targeted: NEFAs, palmitic acid, monounsaturated/polyunsaturated ratio (MUFA/PUFA), linoleic acid, and unsaturated/saturated fat	Linoleic acid was positively correlated with total insulin secretion (*p* = 0.03). Glucose sensitivity correlated with palmitic acid (*p* = 0.01). GBP, SG, and VLCD had similar acute effects that decreased palmitic acid (*p* < 0.05). Several NEFAs correlated with beta-cell function parameters and HOMA-IR (*p* < 0.05).
Luo et al. (2016) [56]	RYGB	35 individuals with T2D, 23 remission and 12 nonremission patients with T2D, were measured at baseline and 6 and 12 months after RYGB.	6 and 12 months	Untargeted: free fatty acids (FFAs), acylcarnitines, amino acids, bile acids, and lipid species	Insulin sensitivity, energy metabolism, and inflammation were related to metabolic alterations in free fatty acids (FFAs), acylcarnitines, amino acids, bile acids, and lipids species (*p* < 0.05). Baseline levels of tryptophan, bilirubin, and indoxyl sulfate measured before surgery as well as levels of FFA 16:0, FFA 18:3, FFA 17:2, and hippuric acid measured 6 months after surgery best predicted the suitability and efficacy of RYGB for patients with T2DM.

CTRL, control subjects; RYGP, Roux-en-Y gastric bypass; T2D, type 2 diabetes; SAT, subcutaneous adipose tissue; VAT, visceral adipose tissue; LSG, laparoscopic sleeve gastrectomy; LAGB, laparoscopic adjustable gastric banding; BPD, biliopancreatic diversion with duodenal switch; DJBL, duodenojejunal bypass liners; GB, gastric bypass; VLCD, very-low-calorie diet; HILIC, hydrophilic interaction liquid chromatography; UPLC–MS, ultra-performance liquid chromatography–mass spectrometry; UHPLC-MS/MS, ultrahigh-performance liquid chromatography–tandem mass spectrometry; 1H NMR, proton nuclear magnetic resonance; LC-MS/MS, Liquid Chromatography with tandem mass spectrometry; VLDL, very low density lipoprotein; LDL, low-density lipoprotein.

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
