# Peer review of "Mechanisms and Outcomes of Metabolic Surgery in Type 2 Diabetes"

_metabolites, 2022, doi:10.3390/metabo12111134_

Round 1

Reviewer 1 Report

The manuscript submitted by Dr. Mansor et al. summarized the current knowledge about the mechanisms and outcomes of metabolic surgery in treating type 2 diabetes. The subject is scientifically sound; however, there are some significant concerns needed to be addressed:

1.       The overall structure of the article has a big problem. The title is "Mechanisms and outcomes of metabolic surgery in type 2 diabetes", so the authors have to discuss all the factors reported in this subject. For example, the modulated gut microbiota is an essential factor, but the authors didn't discuss it in detail. Gut microbiota and the bile acid cross-talk is also needed to be included.

2.       Although the authors searched the literature comprehensively, the discussion of the cited report was very superficial. For example, the authors mentioned, "The mechanisms underlying the benefits of metabolic surgery likely involve the bile acid–signaling pathway [14]", but there is no detail or discussion.

3.       How about weight loss or energy balance-related factors?

4.       Not all diabetes patients are suitable for receiving metabolic surgery. The authors have to introduce the risk-benefit ratio and the limitation of this surgery.

5.       The Abstract is highly similar to the Conclusions section. Why do the authors only emphasize the role of miRNA in the two Sections?

6.       Section 3 should be moved up to Section 2.

7.       Section 4: The subtitle should be divided into "4.1. Hormonal changes" and "4.2. Adipokines changes." The authors should simplify the description of the adipokines.

8.       The whole manuscript's performance has to be improved. The authors are recommended to check the format and font carefully.

Reviewer 2 Report

I think the manuscript is well prepared, it is easily understood. The abstract summarizes the presented work well and can be accepted. I found some mistakes that can be fixed, are the following:

line 81. the numbering of the paragraphs has a mistake.

lines 356 and 357 the word "carbohydrates" is repeated.

Table 1.: I don´t understand the order, it seems to me that it can be arranged possibly by year of publication, intervention or source, in the case that it is chosen, I think it should go in the first column.

Author Response

Response 1:

Thank you for your constructive comments.

The numbering mistakes have been corrected. Table 1 (now table 2) has been arranged according to the year (latest on top). A new table 1 has been added, to support what is recommended by reviewer #1. One of the words "carbohydrates" has been deleted.

Reviewer 3 Report

I think it is interesting review and worth publishing.

But a slight postscript is needed.

Regarding the association of RYGB with GLP-1, author should mention the fecal microbiota and short chain fatty acid (SCFA). An animal study revealed that changed microbiota by RYGB plays a key role in post-surgical weight loss (Cell Metab 2015;22:228–238). Besides bile acids, SCFA was reported to modify the secretion GLP-1 (Physiol Rev 2020;100:171–210). Likely to RYBG in point of inclusion of bypass of the proximal intestine, elevated GLP-1 secretion, altered gut microbiota, and increased SCFA in gut after pancreatoduodenectomy was reported (Diabetes Care 2021;44:1002–1011). Given these, author should describe the series of metabolic changes (gut microbiota and SCFA) derived by bypass of the proximal intestine.

Author Response

Response 1: Thank you for your positive comments.

This part has been added to the hormone section (GLP-1) and discussed in 3.1.b. A new section “3.3 Role of gut microbiota, bile acids, and their cross-talk” and the latest references have also been added.

Round 2

Reviewer 1 Report

The authors have revised the manuscript appropriately according to the reviewer's comments. The manuscript can be accepted for publication.